# Estimating intraclonal heterogeneity and subpopulation changes from bulk expression profiles in CMap

Chiao-Yu Hsieh, Ching-Chih Tu, Jui-Hung Hung 🅾

The connectivity among signatures upon perturbations curated in the CMap library provides a valuable resource for understanding therapeutic pathways and biological processes associated with the drugs and diseases. However, because of the nature of bulk-level expression profiling by the L1000 assay, intraclonal heterogeneity and subpopulation compositional change that could contribute to the responses to perturbations are largely neglected, hampering the interpretability and reproducibility of the connections. In this work, we proposed a computational framework, Premnas, to estimate the abundance of undetermined subpopulations from L1000 profiles in CMap directly according to an ad hoc subpopulation representation learned from a well-normalized batch of single-cell RNA-seq datasets by the archetypal analysis. By recovering the information of subpopulation changes upon perturbation, the potentials of drug-resistant/susceptible subpopulations with CMap L1000 were further explored and examined. The proposed framework enables a new perspective to understand the connectivity among cellular signatures and expands the scope of the CMAP and other similar perturbation datasets limited by the bulk profiling technology.

## Introduction

Connectivity Map (CMap [Lamb et al, 2006]) is a large-scale and comprehensive perturbation database that curates differentially expressed (DE) genes upon diverse perturbagen (i.e., chemical or genetic reagent) treatments in human cell lines. The DE genes induced by each perturbagen represent the perturbed biological pathways that are collectively regarded as a signature. One typical application of CMap is to compare the similarity between a signature and a disease-defining gene list to suggest a positive or negative connection between the perturbagen and disease. Recently, the Library of Integrated Network-based Cellular Signatures (LINCS) project leveraged the L1000 profiling platform, a low-cost and high-throughput profiling technology, to significantly populate the CMap database and offer immense opportunities to new

therapeutics (Wang et al, 2016; Subramanian et al, 2017; Musa et al, 2018).

One founding premise of making sense of the signature from the bulk expression profiling like L1000 is that the clonal cells used for experiments are genetically homogenous so that the signature can reflect the consistent response across cells treated by the same perturbagen. However, in fact, the genetic heterogeneity within human cell lines (e.g., MCF-7 and HeLa) has been confirmed and widely recognized (Fasterius and Al-Khalili Szigyarto, 2018; Ben-David et al, 2019; Liu et al, 2019). Those undetermined subclonal cells bearing distinct genetic variants (i.e., subpopulations) may behave differently upon a perturbation, thereby jeopardizing the interpretability (Laverdière et al, 2018) and reproducibility (Edris et al, 2012; Ben-David et al, 2019) of the signatures by bulk profiling.

The single-cell RNA sequencing (scRNA-seq) technology that combines single-cell isolation and RNA sequencing technologies to study the transcriptome of a single cell enables us to understand the effect of intraclonal/intratumoral heterogeneity ignored in the bulk expression profiling (Chen et al, 2018; Fan et al, 2020). For instance, Ben-David et al (2018) used scRNA-seq to show that the intraclonal heterogeneity in MCF-7 cells may influence the drug response to a great extent. The presence of drug-resistant subpopulations was revealed in MCF-7 cells (Hong et al, 2019) at single-cell resolution. These findings bolster the notion that the signature by bulk profiling cannot be explained solely by pathway perturbation; however, conducting single-cell level assays on the same scale to remedy CMap L1000 datasets in this regard is clearly not realistic.

Recently, digital cytometry approaches (Aran et al, 2017; Newman et al, 2019; Wang et al, 2019; Jew et al, 2020), which use machine learning methods to decompose the bulk gene expression profiles (GEPs) of a heterogeneous cellular mixture (e.g., PBMCs, whole brain tissues, or tumors) into several well-characterized cell types have been proved to be capable of estimating the cellular composition computationally in high accuracy, thereby mitigating the need of conducting single-cell level assays. Despite these powerful digital cytometry approaches, applying them to decomposing bulk GEPs into undetermined subpopulations remains challenging because of the lack of known characteristics of subpopulations of a human cell line. The gaps toward a practical digital cytometry that can recover

Department of Computer Science, College of Computer Science, National Yang Ming Chiao Tung University, Hsinchu, Taiwan

Correspondence: juihunghung@gmail.com

the intraclonal heterogeneity beneath the bulk GEPs by L1000 remain to be filled.

We therefore developed Premnas, a computational framework that first learns the ad hoc subpopulation characteristics from a well-normalized batch of single-cell GEPs via the archetypal analysis (i.e., ACTIONet [Mohammadi et al, 2020]) and then by which estimates the composition of subpopulations from L1000 profiles in CMap using digital cytometry. After recovering the subpopulation composition from each bulk GEP, the change of subpopulation composition upon perturbation can be inferred. The potentials of searching for drug cocktails and drug-resistant subpopulations with LINCS L1000 CMap were further explored and examined. To our best knowledge, this work is the first attempt to provide a new subpopulation perspective to CMap database. We believe Premnas can be applied to all the perturbation datasets, of which intraclonal/intratumoral heterogeneity was concealed by the bulk profiling and hereafter provides a new dimension of interpreting the connectivity.

# Results

## Framework overview

### Rationale

One of the key premises to make use of CMap is that a gene signature, an aggregate of DE genes induced by a perturbagen or disease, can be regarded as the surrogate for the affected functions or pathways. However, because there are subpopulations in a clone, and each subpopulation bears distinct genetic variants and GEPs, fluctuation of the distribution of subpopulations can also account for the gene signature (Fig 1). For instance, if some major subpopulation excessively expressing pathway 1 is highly susceptive of and massively killed by a drug, the genes involved in pathway 1 are easily identified as the negative DE genes upon treatment using bulk GEPs and then regarded as the signature of the drug response. In other words, a gene signature can be a mixed consequence of function and subpopulation changes, especially for the perturbagens that are meant to kill cancer cells.

Because of the nature of bulk profiling, the subpopulation information is unavailable in CMap. The conventional drug screening strategies that interpret gene signatures and connections without considering possible compositional change could jeopardize the conclusions drawn. For instance, cancer drugs suggested by CMap may be deemed ineffective and necessitate further investigation to increase the reproducibility (Ben-David et al, 2019) because of the underlying composition bias in samples. The goal of our framework, Premnas, is meant to enable the CMap to interpret gene signatures at both the functional and subpopulation levels. The workflow of Premnas is illustrated in Fig 2 and explained below.

To begin with, the first difficulty to tackle was the unknown characteristics of each subpopulation. We approached this issue by making the following assumptions:

**Assumption 1** There are a bounded number of subpopulations universally within a cell line. That is, most of the representative

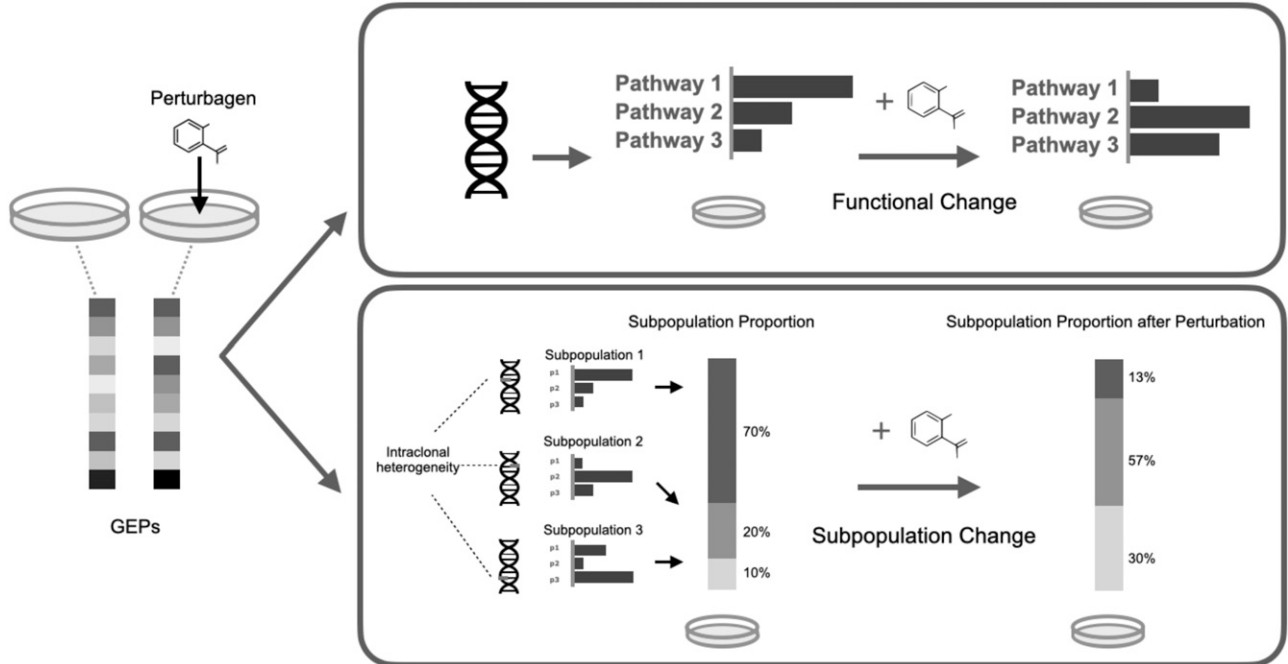

**Figure 1.   Changes of gene expression profiles upon a perturbagen could be a mixed consequence of function and subpopulation changes.**
(Top) The conventional perspective regards gene signatures as the perturbed pathways of a homogenous cell clone. (Bottom) In a heterogenous clone, each subpopulation bearing distinct genetic variations drives various pathways and has different susceptibility to the perturbagen. The gene signature therefore reflects the change of intraclonal heterogeneity.

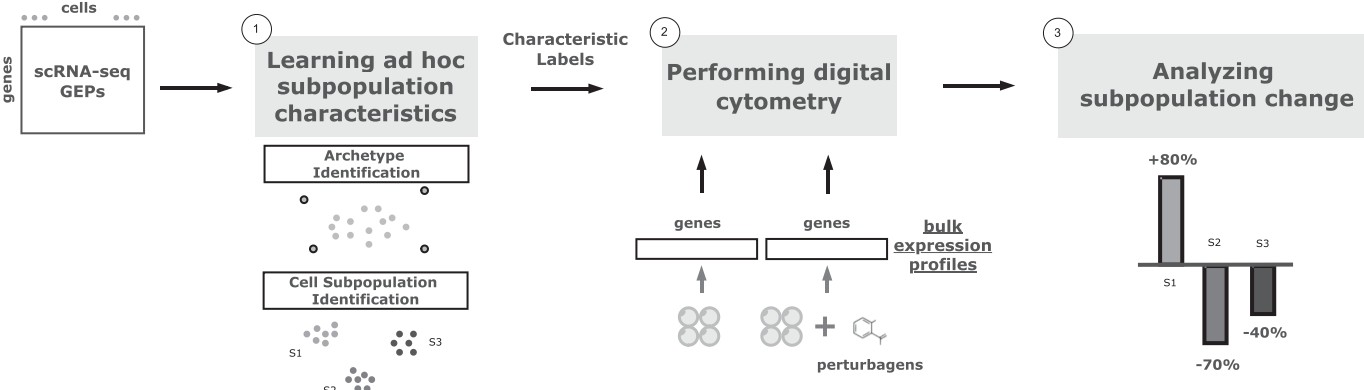

**Figure 2. The scheme of Premnas.**
First, single-cell gene expression profiles are used as input of archetypal analysis. The subpopulation characteristics could be learned, and all the cells would be labeled with its belonging subpopulation. Then a digital cytometry is performed with bulk expression profiles (bulk with and without perturbagens are both used) and enable us to estimate cell subpopulation abundances. Finally, the subpopulation change upon a perturbation would be calculated, and the effect of the perturbagens on each subpopulation could be further examined.

subpopulations of a human cancer cell line should be present in a large enough pool of scRNA-seq datasets collected from different sources.

**Assumption 2** Cells of the same subpopulation should collectively share invariant subpopulation characteristics, and each subpopulation can be distinguished by its unique subpopulation characteristics despite perturbations.

### Learning ad hoc subpopulation characteristics

With the above assumptions, intuitively, subpopulation characteristics can be learned from a pooled scRNA-seq data by dimension reduction approaches, such as nonnegative matrix factorization (NMF [Lee & Seung, 1999]), t-distributed stochastic neighbor embedding (t-SNE) (van der Maaten, Laurens and Hinton), and UMAP (McInnes & Healy, 2018), accompanied by some clustering methods (Puram et al, 2017; Gan et al, 2018) to identify subpopulations. Yet, nonlinear approaches like t-SNE and UMAP obscure the biological interpretation of subpopulation characteristics, whereas the traditional NMF algorithm tend to omit weakly expressed but highly specific cell states.

We decided to use ACTIONet (Mohammadi et al, 2020), a tool designed specifically for subtyping cells with scRNA-seq, to ensure biological interpretability during dimension reduction. The concept of ACTIONet is similar to NMF; however, it directly distills the most representative cell states (termed "archetypes") from the single-cell GEPs of multiple samples and groups cells into subpopulation in the archetypal-based metric cell space. In addition, to make sure that ACTIONet does not recognize technical and biological noises (e.g., batch effects and cell cycle-related functions, respectively) as archetypes, such differences are removed by the embedding-based normalization (i.e., Harmony [Korsunsky et al, 2019]) before performing the archetypal analysis. Besides, ACTIONet does not need prior knowledge of the number of underlying archetypes as required in traditional NMF; instead, it conducts different decomposition levels to ensure the robustness of finding archetypes. After cell subpopulations were identified by ACTIONet, we pruned the

nonrepresentative cells and derived the subpopulation characteristics for each subpopulation (see the Materials and Methods section).

### Performing digital cytometry

Once the underlying subpopulations were identified, the most straightforward way to estimate their abundance in bulk samples is by conducting a simple linear regression to model the relationship between the bulk GEP and subpopulation characteristics. However, integrating subpopulation information into the CMap database was nontrivial because of the considerable technical variation between the different profiling technologies (e.g., scRNA-seq and L1000). CIBERSORTx (Newman et al, 2019) is capable of adjusting the matrix of subpopulation characteristics derived from the scRNA-seq GEPs while decomposing the query bulk GEPs into the distribution of cell subpopulations with support vector regression. Thus, after pre-processing and normalizing GEPs from scRNA-seq and CMap, we performed the digital cytometry by CIBERSORTx to assess the subpopulation distribution in each experimented sample from the CMap database (see the Materials and Methods section).

### Validation

Because of the lack of known gene markers of subpopulations in cancer cell lines, we were unable to find data from studies that performed flow cytometry to label the identity of each cell accompanied by matched GEP profiles for validation. We then relaxed our criteria and collected data on PBMCs to serve our purpose. We used the same scRNA-seq and bulk RNA-seq datasets of PBMCs as in the original paper of CIBERSORTx (Newman et al, 2019) to test the validness of the proposed workflow (see PBMC verification in Supplemental Data 1). Through Premnas, we found nine subpopulations among PBMCs (See Fig S1A and B), annotated their cell type by known marker genes, and estimated their abundance in the bulk samples. The Pearson correlation coefficient between the composition estimations via the digital cytometry based on the ad hoc subpopulation characteristics and the ground truth composition directly assessed by flow cytometry was high (r = 0.835) (see Fig S1C

and D). Moreover, in addition to the bulk RNA-seq, we also performed the deconvolution validation on the microarray (see Fig S2) platform. The estimation based on microarray also showed a high correlation with the ground truth (r = 0.80 by Pearson correlation coefficient). These results suggest that Premnas can discover the unspecified subpopulation from scRNA-seq data and estimate the distribution of cell subpopulations in bulk samples correctly.

### Analyzing subpopulation changes

After getting the abundance distribution of subpopulations in bulk GEPs, the intraclonal heterogeneity can be estimated (e.g., by Shannon's entropy), and the changes between distributions under different conditions (e.g., between control and perturbed samples) can further reveal the effects of a treatment to a specific subpopulation. For instance, subpopulations that are either more resistant or susceptible to a specific drug at a particular concentration can be identified. Moreover, the biological functions of these subpopulations can be explained by their underlying archetypes.

### Applying Premnas to the LINCS L1000 CMap library

There were 1.3 million bulk GEPs (2,710 perturbagens, 3 time points, 26 cell lines, and 117 concentrations) available in the LINCS L1000 CMap library. MCF-7 based GEPs constituted the most comprehensive collection (39,711 GEPs for 1,761 perturbagens), and recent research had discovered MCF-7 subpopulations through single-cell technologies (Hong et al, 2019; Muciño-Olmos et al, 2020), which made MCF-7 a feasible cell line for the demonstration of Premnas. Of note, the biological noises in scRNA-seq data that could dampen clustering accuracy, including cell cycle effects and clonal differences, were carefully examined and reduced by a series of preprocessing procedures (See the Materials and Methods section and Figs S3 and S4).

### Identification and validation of MCF-7 subpopulations learned from scRNA-seq datasets

After the ad hoc subpopulation characteristics learning step in Premnas, 10 subpopulations (Fig 3A), which consist of 17 archetypes, were identified (See the Materials and Methods section). Each of the 17 archetypes possessed unique highly expressed genes as assumed in Assumption 2 section (see Figs 3B and S5). We then performed the enrichment analysis to understand the characteristics of each subpopulation in MCF-7. Gene ontology and gene set enrichment analysis were then conducted with Metascape (Zhou et al, 2019). After pruning (see the Materials and Methods section), every cell had a major archetype and a subpopulation identifier. The composition of the main archetypes of each subpopulation and the top 3 significant pathways (ranked by the q-values calculated by Metascape) in each archetype can be found in Fig S6.

To assure that the 10 subpopulations were comprehensive enough as stated in the Assumption 1 section, we used the scRNA-seq datasets (Hong et al, 2019), in which an MCF-7 cell subpopulation (i.e., preadapted cells; PA cells) showing resistance against drugs after endocrine therapy was identified, to see whether any of the 10 subpopulations resembles PA cells. We colored the MCF-7 cells used for the previous subpopulation identification based on the expression of the two reported marker genes of PA cells (i.e.,

CD44 and CLDN1) and discovered that most of the cells expressing a higher degree of these marker genes tended to aggregate in subpopulation 2, 4, and 9 in the UMAP plot (Fig S7).

Furthermore, we reran Premnas on the merged MCF-7 dataset, including the datasets used for subpopulation identification above (GSE114459 [Ben-David et al, 2018]) and the ones treated with endocrine therapy (GSE122743 [Hong et al, 2019]) and see whether any of our previous found subpopulations can be grouped with known PA cells (see Fig 3C). Likewise, biological and technical noises were eliminated in advance. Note that the cell pruning was skipped for a more comprehensive comparison. Premnas identified 12 clusters from the merged dataset and showed that 53 of 81 PA cells (63.5%) were assigned to cluster 2. Moreover, only the number of cells in cluster 2 showed a constant increase in the datasets with the longer endocrine treatment (i.e., 4 and 7 d; see Fig 3D and E). In addition, cells from GSE114459 in cluster 2 were originally annotated as subpopulation 2 (See Fig S8). Based on the evidence, we believed that PA cells were mostly covered by the subpopulation 2. The enriched pathways linked to subpopulation 2 also help explain the drug-resistance of PA cells (see below). Although this was just one example, it is still an indication that the 10 subpopulations indeed cover cells that was not present in the training data, supporting the Assumption 1 section. Note that as more and more scRNA-seq datasets are getting available, the subpopulation characteristics can be retrained on the pooled datasets and further improve Premnas' sensitivity in subpopulation identification.

### Drug-susceptible subpopulations inferred from bulk GEPs reflects drug-induced pathways

With the subpopulation characteristics of MCF-7, we tested whether the perturbed subpopulations found by Premnas complied with known facts before applying Premnas to measure the subpopulation changes in all the bulk GEPs in LINCS L1000. We used Premnas to decompose 12 bulk GEPs of MCF-7 treated with FDI-6 (GSE58626 [Gormally et al, 2014]), in which the experiments were designed to assess the FDI-6 effects on MCF-7 by RNA-seq in triplicates at different time points (0, 3, 6, and 9 h). FDI-6 has been known for repressing the growth of MCF-7 cells. We compared the distributions of subpopulations from controls with those from treated samples to determine the affected subpopulations. FDI-6, which is known for displacing FOXM1 (Gormally et al, 2014), is an important mitotic player that involved in cancer progression and drug resistance in MCF-7 cells (Ziegler et al, 2019) and induces coordinated transcription down-regulation.

The relative changes in cellular composition after FDI-6 treatment were estimated and shown in Fig S9A and B. Both subpopulation 6 and 7 were completely inhibited after treatment; however, FDI-6 had the most significant impact on subpopulation 6 by reducing its abundance from 18% of all cells to 0%. The characteristics of subpopulation 6 and 7 were explained by their main archetypes (i.e., archetype 16 and 14, respectively), which were associated with mitotic processes, cell cycle regulation and so on (Fig S9C).

The major functional features of the perturbed subpopulations concurring with the known roles of FOXM1 as a key regulator of M phase progression and cell cycle regulation (Ziegler et al, 2019) indicated that the subpopulation distinguished by the targeted pathways were more susceptible to the FDI-6. The result also

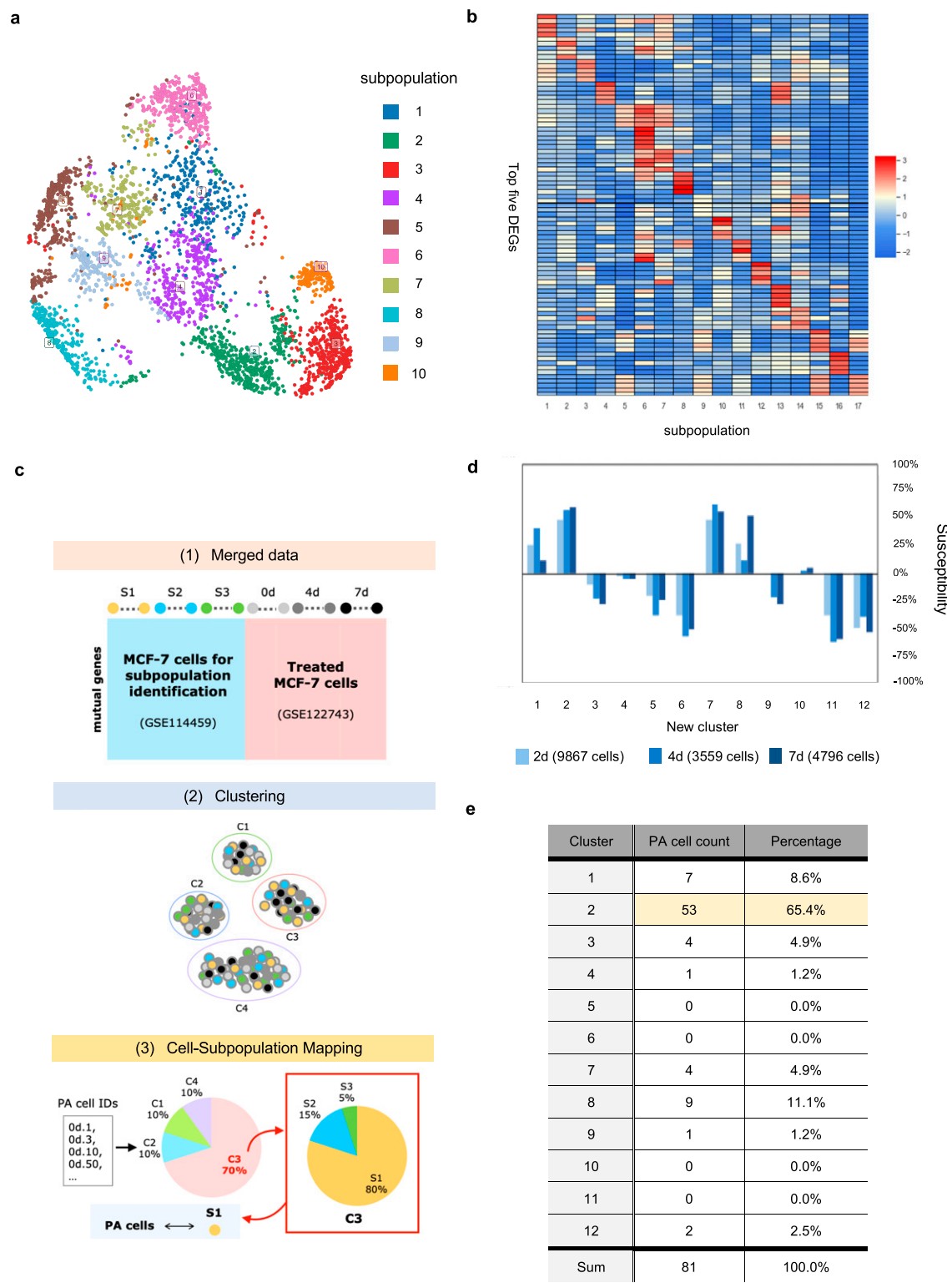

**Figure 3.  MCF-7 subpopulation and PA cell identification.**
**(A)** UMAP visualization of GSE114459 cells after pruning the cells with inexplicit archetype representation. Ten subpopulations were identified by ACTIONet. **(B)** The expression profile of the top five DEG in each archetype. The rows of the heatmap were normalized by z-score normalization. The unique expression of these genes across archetypes implies the functions represented by 17 archetypes. **(C)** An illustration of the process of the PA cell comparison. We first merged two datasets (GSE114459 and GSE122743) and clustered the cells by ACTIONet. PA cell IDs were then used to recognize those PA cells in each cluster. Finally, by mapping the new clusters to the old subpopulations, we could identify the subpopulation covering the most PA cells. **(D)** Changes in the distribution of GSE122743 cells with different treatment durations in the new clusters. **(E)** PA cells distribution among 12 clusters in the merged data. Cluster 2 had been found including the most PA cells among all of the clusters.

demonstrated that Premnas can be used to study drug effects in the perspective of both the intraclonal heterogeneity change and biological functions.

### Identification of the most drug-resistant cell subpopulation in MCF-7 from LINCS L1000 bulk GEPs

We then set out to apply Premnas to LNCS L1000 CMap datasets. With 39,710 (1,760 perturbagens, 107 different concentrations ranged from 0.004 to 20 $\mu$M, and three time 3, 6, 24 h) MCF-7 GEPs downloaded from the GEO website (GSE70138, version: 2017-03-16) as input, we found that there were many perturbagens that caused great reduction of intraclonal heterogeneity. To better delineate drug effects on inhibiting the growth of MCF-7 cells in the subpopulation perspective, we defined two metrics: drug susceptibility and treatment consistency. The drug susceptibility of a cell subpopulation, which ranged from −100 to 100%, was defined by its relative change in proportion after treatment. The consistency was calculated as the median drug susceptibility for experiments using the same drug but at a higher dose. This study considered a cell subpopulation with a susceptibility less than −90% after treatment as highly drug-susceptible (or say, killed) by the drug.

We calculated the drug susceptibility and treatment consistency for every perturbagen–concentration–time pair (PCT pair) of LINCS L1000 MCF-7 datasets and tried to find drug-resistant subpopulations. Surprisingly, among 1,760 unique perturbagens in the LINCS CMap database, subpopulation 2 can survive in all PCT pairs. Interestingly, subpopulation 2 is also what we found representing the drug-resistant PA cells from the endocrine therapy (Hong et al, 2019) datasets (GSE122743; see Fig 3).

To further understand the causes of the drug resistance, we looked into the characteristics of archetype 5, the primary archetype of subpopulation 2. Enriched functions of archetype 5 were involved in transforming growth factor $\beta$ receptor signaling pathway and extracellular matrix organization (Fig S6). This result coincided with the previous studies that stated an essential role of TGF-$\beta$ in drug resistance in cancer (Brunen et al, 2013). Many of the top DE genes of archetype 5 (see MCF-7 DEGs in Supplemental Data 1), including GPRC5A, ITGAV, SEMA3C, and ITGB6, have been proven to associate with breast cancer susceptibility to apoptosis or treatment and poor prognosis (Moore et al, 2014; Zhou & Rigoutsos, 2014; Malik et al, 2016; Cheuk et al, 2020).

The facts that no drug used in CMap can effectively kill cells of subpopulation 2 and that the known, drug-resistant PA cells are enriched in subpopulation 2 suggest that PA cells might be a valuable research targets for understanding the drug resistance of breast cancer cells, and more efforts should be focused on designing drug targeting PA cells.

## Discussion

After getting the drug susceptibility and treatment consistency of all PCT pairs of LINCS L1000 MCF-7, we came up with a greedy search strategy for suggesting a minimal therapeutics combination (i.e., a cocktail therapy) by aggregating perturbagens that kill specific subpopulations, where no subpopulation could survive after the treatment.

The strategy (Fig 4) begins with calculating the susceptibility of each subpopulation for every perturbagen–concentration–time pair (PCT pair) of LINCS L1000 MCF-7 datasets and then iteratively selecting a PCT pair that can kill the greatest number of subpopulations. The perturbagen of the pair should also present with the high consistency (~80%) across higher doses, and the PCT pair with the lowest concentration is added to the cocktail. The killed subpopulations and all PCT pairs linked to the selected drugs are removed from the search. The iteration continues until no more subpopulation could be killed. See the Materials and Methods section for more details.

After searching among all the PCT pairs with our greedy search strategy, four PCT pairs were chosen as a potential drug cocktail: 3.33 $\mu$M A-44365 for 24 h, 0.12 $\mu$M UNC-0638 for 24 h, 0.041 $\mu$M gemcitabine for 24 h, and 0.123 $\mu$M ixazomib-citrate for 24 h. Nine of 10 MCF-7 cell subpopulations could be killed by the cocktail (Fig 5A) and the susceptivity strengthened along with higher dosage (Fig 5B). With the subpopulation change estimated by Premnas, our strategy can be used to suggest drug cocktails for potently suppressing breast tumor cells that share a similar genetic background with the MCF-7 cell line.

We did not carry out further experiments to verify the effectiveness of the drug cocktail, but there are many studies that have already proved the antitumor activities of each selected compound, supporting the feasibility of this treatment combination. For instance, UNC-0638, an inhibitor of G9a and GLP, was reported to exert inhibitory effects against MCF-7 cells (Vedadi et al, 2011). G9a is known to participate in hypoxia response in MCF-7 cells (Riahi et al, 2021), whereas subpopulation 10, the target subpopulation of UNC-0638 in the treatment selection process, is also associated with oxidative phosphorylation. Moreover, gemcitabine, another perturbation we chose, had also been demonstrated to be sensitive with mRNA expression levels of some genes (Meng et al, 2015), consistent with the result in our studies that the main pathway of the best-killed subpopulation of gemcitabine is the regulation of mRNA metabolic process. Based on these studies, we believed the therapeutic combination would exhibit potent antitumor activity with partially increased doses in MCF-7 cells. Issues such as drug interactions (e.g., synergy or antagonism) were clearly crucial but omitted in the search strategy, and more experiments have to be conducted in the future to improve the search strategy.

In the development of Premnas, we found that careful preprocessing to remove technical and biological biases and noises among all single-cell GEPs before performing the learning step of the subpopulation characteristics was of great importance. Normalization steps (e.g., quantile normalization, Harmony, etc.) were helpful, but our experience suggests that some datasets should be carefully examined, adjusted, or even removed from the training data if they lead to some obvious isolated, distant subpopulations when projecting to the embedding space. The enriched pathways of the major archetypes associated with the subpopulations should also be scrutinized to make sure those subpopulations are meaningful.

The precise recognition of subpopulations also relies on the comprehensiveness of the collected scRNA-seq profiles of the cell line. Because the MCF-7 clones used in this study were single-cell–derived from the same parental clone, it increased the

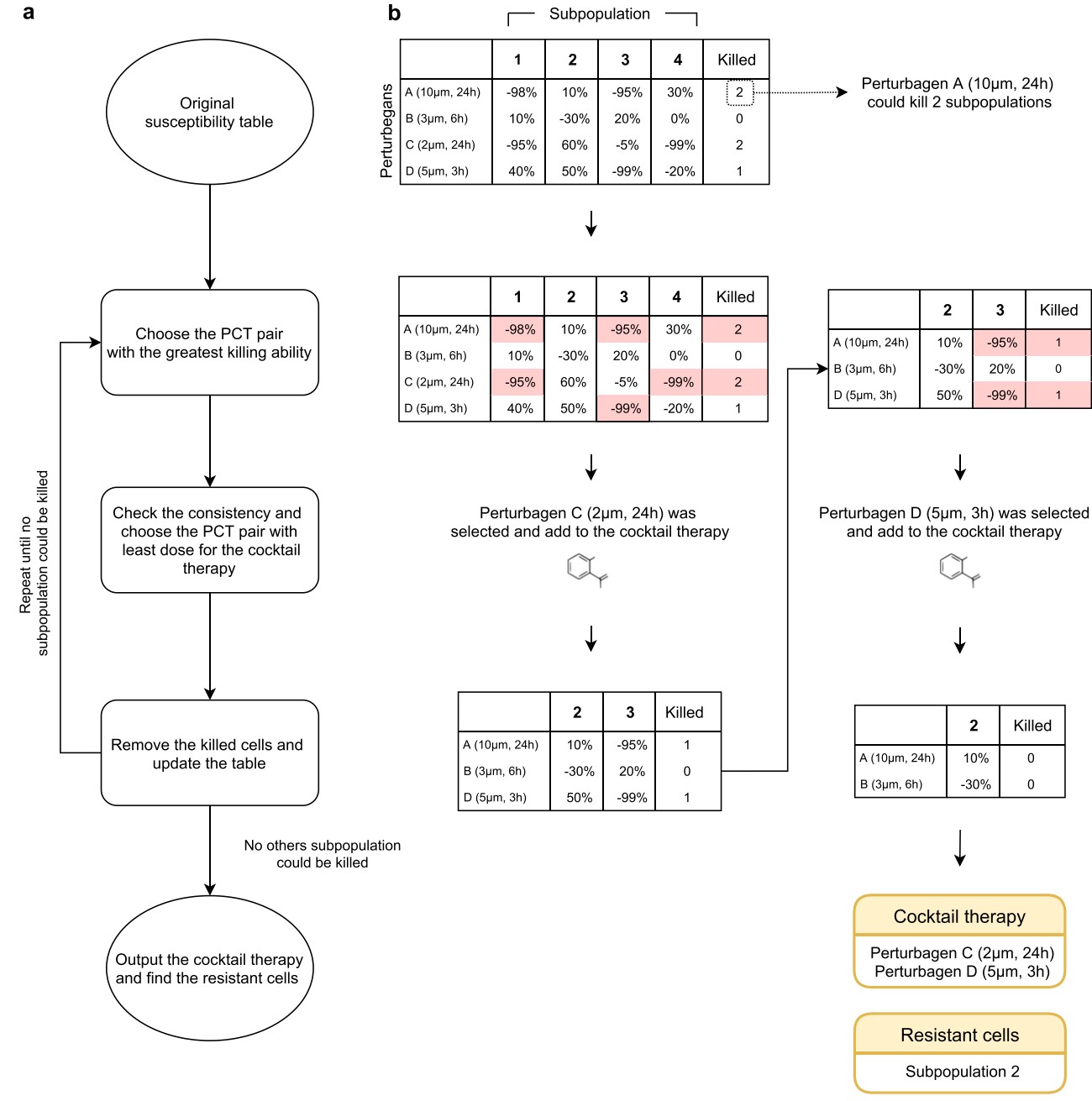

**Figure 4. An illustration of the greedy search for suggesting cocktails.**
**(A)** The workflow. The susceptibilities of each subpopulation are first evaluated for every PCT pair, and the full susceptibility table is constructed. Then the most lethal PCT pairs, which could kill the most subpopulations, are chosen if their consistencies are deemed high. If there is more than one PCT pair, the PCT pair with the least dose is included in the cocktail therapy. The selection repeats until no more subpopulation could be killed. **(B)** A simple example of the greedy search. Perturbation C and perturbation D are eventually added to the cocktail therapy.

probability of failing to capture all the possible genetic evolution of MCF-7 cells. For instance, when we included the scRNA-seq datasets for the cells from the endocrine therapy (Hong et al, 2019) datasets (GSE122743), two new subpopulations were reported. Including as many single-cell transcriptomic data of the cell line of interest for a more comprehensive analysis should be taken for all further research applying Premnas.

The differences between profiling technologies place a difficulty in estimating subpopulation distribution in bulk samples. CIBER-SORTx (S-mode) reduced the technical variation in gene expression by using an artificial mixture to help tune the signature matrix (see the Materials and Methods section). Furthermore, the bulk GEPs we encountered was largely conducted by the L1000 and RNA-seq, and they were designed to quantify different gene sets. That is, it is

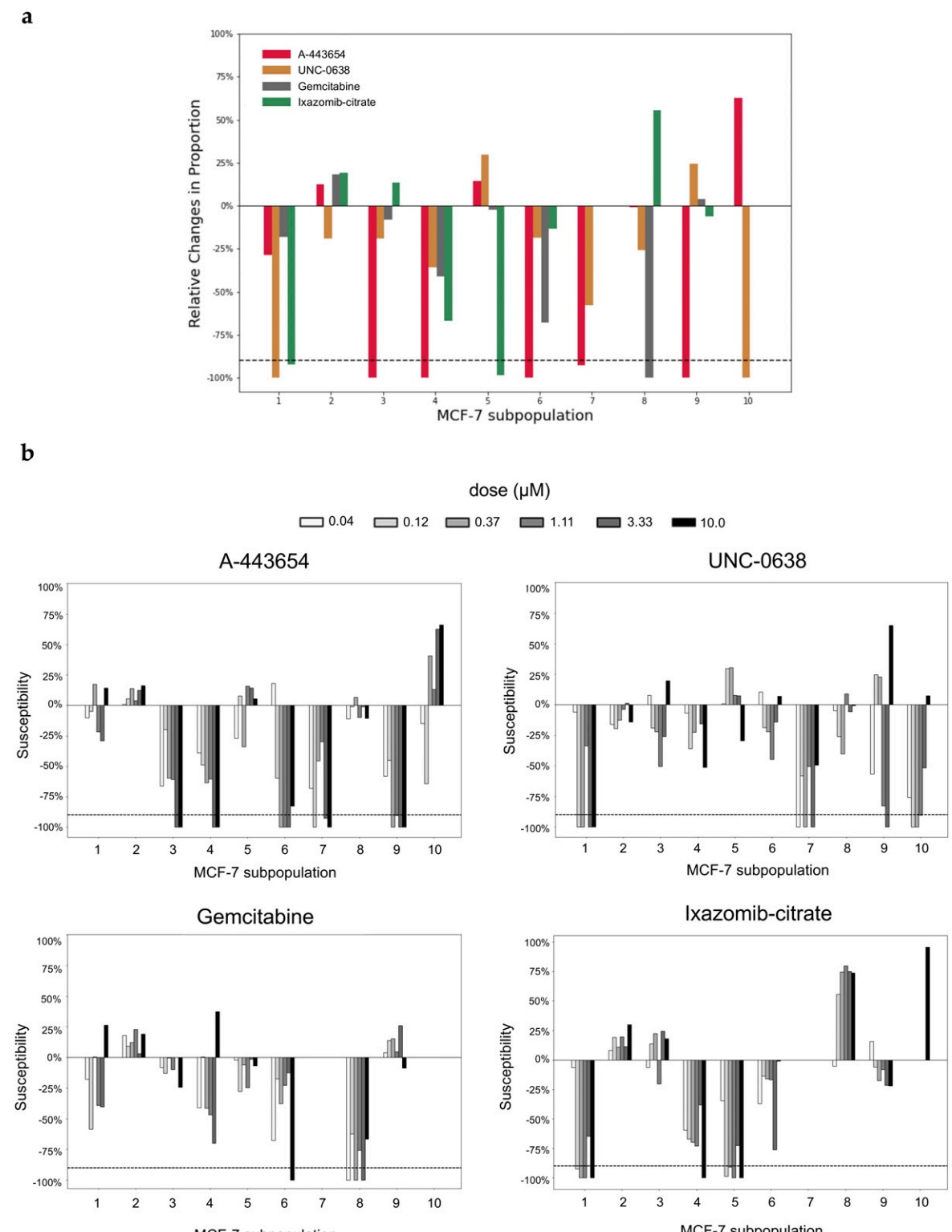

**Figure 5. The effects of selected perturbagens in the cocktails.**
**(A)** The average relative changes in cell subpopulations after perturbations. MCF-7 was treated with 3.33 μM A-443654 for 24 h, 0.12 μM UNC-0638 for 24 h, 0.041 μM gemcitabine for 24 h, and 0.123 μM ixazomib-citrate for 24 h in replicates. The dotted line represents the −90% threshold of high susceptibility. **(B)** Selected perturbations showing dose-dependent effects. The duration of perturbations shown is 24 h, and the doses are in micrometer.

possible that some genes involved in the learning of subpopulation characteristics do not present in bulk GEPs. Because CIBERSORTx is a marker gene-based decomposition approach, the calculation could depend on some of those missing genes, thereby compromising accuracy.

We think Premnas can be applied to all kind of perturbation-based bulk GEP datasets to understand the effect of the perturbagens to the distribution of uncharacterized subpopulation within a cell line or tumor tissue sample. In addition, it might be worth trying to use Premnas for checking the intraclonal heterogeneity of the controlled samples. If a controlled sample shows a biased subpopulation composition, extra cautions should be taken to assure the genetic background of the cells used before further analysis or comparison, which may be helpful to the reproducibility of the experiments.

The logical basis of Premnas relies on the assumption that there are invariant subpopulation characteristics to represent each subpopulation so that the fluctuation of expression of these subpopulation characteristics can be solely explained by changes in subpopulation composition. However, in practice, the inferred gene signatures can be the mixed consequences of the subpopulation and function changes, therefore violating the assumption. As a result, it is possible that the subpopulation changes reported by Premnas can be because of cells changing their behaviors and acting like some other subpopulations upon a treatment. Unfortunately, it is pretty unlikely such a difference can be distinguished from the information given in the bulk GEPs in the current setting. It is strongly recommended to always refer to the DE genes or enriched functions associated with the major archetypes of the affected subpopulations and thereby interpret the results also from the function perspective. It is important to keep open to alternative explanations of the results.

## Conclusions

Large-scale perturbation databases, such as LINCS CMap, that use cost-effective bulk profiling assays to reveal signatures upon perturbation, and thereby construct the connectivity between the drugs and diseases that share positive or negative correlation of signatures, are the valuable resource of drug discovery. However, the possibility that the signature is driven by the subpopulation changes is largely unexplored because of the lack of the companion single-cell assays. This study is the first attempt to expand the scope of interpretation and application of the LINCS CMap database in regard of intraclonal cellular composition.

The three main steps of the proposed framework Premnas include (1) learning the ad hoc subpopulation characteristics of cells using single-cell transcriptome data, (2) using the subpopulation information to decompose the bulk GEPs by the digital cytometry approach and estimate the abundance of each subpopulation, and (3) comparing the subpopulation compositions under different conditions to understand the effects of drugs to specific subpopulations.

We applied Premnas to MCF-7 cell line data and identified 10 cell subpopulations. We found consistent experimental evidence to support the classification. After dissecting the effects of thousands

of perturbations on MCF-7 cells from the bulk profiling assays curated in the LINCS CMap, we further discovered the most resistant subpopulation among MCF-7 cells and associated its characteristics to the known PA cells. The result suggested that Premnas can be applied to perturbation datasets to reveal intraclonal/intratumoral heterogeneity and provides a new dimension of interpreting signatures and connectivity.

## Materials and Methods

### Data preprocessing

For scRNA-seq data of MCF-7, cells in GSE114459 and GSE122743 were labeled by their source clones (i.e., parental, WT3, WT4, and WT5) and their treatment duration (i.e., 0, 2, 4, and 7 d). We excluded cells with low quality by the criterion used in the original papers: MCF-7 cells with >15% or <1% mitochondrial content and potential multiplets cells with >5,000 and <1,000 expressed genes were removed; as for PBMCs, cells with >10% or <1% mitochondrial content or >3,500 and <500 expressed genes were removed. A total of 1,054 cells in PBMC data, 12,730 cells in GSE114459, and 28,389 cells in GSE122743 were kept for the downstream analysis. Of note, because the count matrix of GSE122743 did not contain mitochondrial genes, we also removed the genes begin with "MT-" from the GSE114459 dataset when merging these two datasets.

For L1000 data, expression data were $\log_2$-transformed, which is not acceptable by CIBERSORTx, so we transformed the data back to the original space. Probe IDs were mapped to gene names with the information in the file "GSE70138_Broad_LINCS_gene_info_2017-03-06.txt." To ensure the authenticity of computed effects, we only keep the experiment results of perturbations with three or more replicates for analysis in this study.

### Removal of biological or technical noise

Intra-type variation may impair the performance of clustering algorithms by grouping cells with similar status (such as cell cycle or technical bias) together rather than cells with the same cell types. We used the Harmony (Korsunsky et al, 2019) algorithm for removing possible confounding status (or say, noise) among batches of samples, which was included in the ACTIONet package (version 2.0). Harmony takes a PCA embedding and batch assignments of cells as input. In this study, we combined the tags of the source clone and the cycle phase (including the dataset label when merging two MCF-7 datasets) as a batch assignment for individual cells (e.g., "WT3_S," "parental_G1," or "WT5_G2_GSE114459"). The first step in the Harmony algorithm is to compute a fuzzy clustering by using a batch-corrected embedding, whereas ensuring the diversity among batches within each cluster was maximized. Next, the algorithm corrects the batch effects within clusters. These procedures are iterated until the cluster assignment of cells becomes stable. After eliminating the noises from the transcriptome data with the Harmony algorithm, our clustering result was no longer affected by the cell cycle phase and the clone of origin (see Figs S4 and S10A and B).

### Selection of the depth parameter for ACTIONet construction

With Harmony-corrected data, we conducted the archetypal analysis with the function *run.ACTIONet()* in the ACTIONet package. However, like in the original NMF, the degree of resolution determined by "k_max" parameter can directly affect the efficacy of capturing biological information under single-cell transcriptome data (Table S1). We tried eight different values for the k_max parameter and recorded the resulting numbers of archetypes and subpopulations (Table S2). We found that when set k_max to the default value (i.e., 30), ACTIONet identified the most subpopulations (10 subpopulations) with the least number of archetypes (17 archetypes).

### Clustering

The cell clustering was accomplished by the *cluster.ACTIONet* function with the clustering resolution parameter = 1 in the ACTIONet package. ACTIONet transformed the metric cell space into a graph to reduce computational time and used the Leiden algorithm (Traag et al, 2019) to detect communities. To prevent the noise caused by ambiguous cells performing multiple cell states, we pruned the cells by considering their composition of archetypes (i.e., the archetypal explicit function), which would be calculated by ACTIONet and represented the convex combination of archetypes for each cell. Cells with their archetypal explicit function below 0.6 were pruned before the downstream analysis. Results with different pruning threshold of PBMC are shown in Fig S3, and the final pruning results of MCF-7 are in Fig S10C and D. The characteristics of the 10 MCF-7 subpopulations identified by clustering can be elucidated by their most influential archetype afterward.

### Decomposition of bulk GEPs by CIBERSORTx

CIBERSORTx took single-cell reference profiles with cell-type annotations and mixture profiles derived from bulk tissues as inputs. All the GEPs should be normalized into the same scale beforehand for more accurate estimation. In this study, the summation of gene expression for each sample was normalized to one million. In addition to single-cell reference profiles and mixture profiles, the decomposition input also included a signature matrix generated by CIBERSORTx. To construct the signature matrices from the scRNA-seq profiles of MCF-7 cells and PBMCs (Fig S11), the DE genes along cell subpopulation types were identified using a Wilcoxon rank-sum test with *P*-value < 0.01. CIBERSORTx removed the genes with low expression (average 0.5 counts per cell in space) and generated the signature matrices as described previously (Newman et al, 2019). The use of a signature matrix in CIBERSORTx helped facilitate faster computational running time during decomposition because of the reduction of the number of genes. After collecting all the input data, CIBERSORTx was able to decompose the bulk-tissue profiles into proportions of cell types/subpopulation while correcting the variation caused by different sequencing techniques.

To enhance the robustness of the CIBERSORTx output, the permutations for statistical analysis was set to 500 (which could be set as a parameter in CIBERSORTx). Moreover, to eliminate the technical variation between 10X Chromium and bulk, we applied S-mode correction provided by CIBERSORTx to our deconvolution process. We briefly introduce the S-mode strategy here: Given a cell-type–annotated single-cell reference profile matrix (m genes X n single cells) from which the signature matrix (m genes X k cell types) was constructed, CIBERSORTx created an artificial mixture profile (m genes X *P* artificial samples) with a known fraction. After CIBERSORTx corrected the batch effects between and the real mixture profile, the adjusted signature matrix could be computed by the nonnegative least squares algorithm (NNLS), given the adjusted artificial mixture profile and its corresponding fraction. Eventually, CIBERSORTx used the support vector regression algorithm (SVR) to estimate the composition of cell types under the real mixture profile with the adjusted signature matrix. The CIBERSORTx team has shown that the deconvolution performance was significantly improved with the single-cell signature matrix adjusted by S-mode correction in their original paper. We also performed some sampling experiments from the PBMC datasets to examine the robustness of decomposition by CIBERSORTx (see Figs S12 and S13).

### Susceptibility of a perturbagen treatment

We evaluated the inhibitory effects of each perturbation based on susceptibility. The susceptibility of a cell subpopulation, which ranged from –100–100%, was calculated as below.

$$Susceptibility = \frac{\sum_{j \in P} \frac{TC_j - \overline{CC}}{TC_j + \overline{CC}}}{|P|}$$

*P*: replicate indices.

$TC_j$: a vector storing the cell subpopulation composition in the treated sample j measured by CIBERSORTx.

$\overline{CC}$: a vector storing the average composition of cell subpopulations in the control samples from the same detection plates as the treated samples.

## Data Availability

### Premnas

The executable and source code of Premnas is freely available at https://github.com/jhhung/Premnas.

### scRNA-seq data

Three single-cell datasets were used in this study, including two MCF-7 datasets (GSE114459 and GSE122743) and one PBMC dataset (GSE127471). All of the cell count matrices were generated by the 10x Genomics Chromium platform and preprocessed by Cell Ranger (Zheng et al, 2017). Rows of the count matrices were gene names. The wild-type MCF-7 cells collected in the GSE114459 dataset were obtained from three clones (i.e., WT3, WT4, and WT5) and their parental clone and used for subpopulation identification in this work. MCF-7 cells in the GSE122743 dataset were treated with E2 depleted medium. We pooled nine samples (GSM3484476 - GSM3484484) from the GEO website together for PA cell

identification. Reads of MCF-7 cells were aligned to GRCh38 with Cell Ranger v2.1. The PBMC dataset was originally generated to evaluate the decomposition performance of CIBERSORTx and we also used it to validate the Premnas.

### RNA-seq data

The RNA-seq dataset of MCF-7 with FDI-6 treatment was downloaded from the GEO website with the accession number of GSE58626, and it contained the GEPs of MCF-7 cells treated with 40 $\mu$M FDI-6 for 3, 6, or 9 h in triplicates. We applied Salmon (Patro et al, 2017) v1.2.0 for alignment-free transcript quantification with the GRCh38 index set and the default parameters. Ensembl IDs were converted to gene name according to GRCh38 reference.

### L1000 data

The 39,710 quantile-normalized L1000 profiles for MCF-7 in the LINCS CMap database were generated with the three files downloaded from the GEO website ("GSE70138_Broad_LINCS_inst_info_2017-03-06.txt," "GSE70138_Broad_LINCS_Level3_INF_mlr12k_n345976x12328_2017-03-06.gctx," "GSE70138_Broad_LINCS_pert_info_2017-03-06.txt," and "GSE70138_Broad_LINCS_gene_info_2017-03-06.txt").

### Microarray data

The microarray data of PBMCs from 10 humans were downloaded from the GEO website with the accession number GSE106898. The expression data was quantile-normalized, and the probe IDs were transformed into gene names accordingly.

## Supplementary Information

## Acknowledgments

This work was supported by Ministry of Science and Technology (110-2622-8-009-009-TA and 110-2221-E-A49-069-MY3 to JH Hung) of Taiwan.

### Author Contributions

C-Y Hsieh: conceptualization, data curation, software, formal analysis, validation, investigation, visualization, methodology, and writing—original draft.
C-C Tu: data curation, software, validation, investigation, visualization, and writing—original draft.
J-H Hung: conceptualization, investigation, methodology, project administration, and writing—original draft, review, and editing.

### Conflict of Interest Statement

The authors declare that they have no conflict of interest.

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
