## [Reviewer comments · Life Science Alliance]

Life Science Alliance

Estimating Intraclonal Heterogeneity and Subpopulation Changes from Bulk Expression Profiles in CMap

Chiao-Yu Hsieh, Ching-Chih Tu, and Jui-Hung Hung

DOI: <https://doi.org/10.26508/lsa.202000745>

Corresponding author(s): Jui-Hung Hung, National Chiao Tung University

Review Timeline:

Submission Date:	2021-11-16
Editorial Decision:	2022-01-24
Revision Received:	2022-04-24
Editorial Decision:	2022-05-18
Revision Received:	2022-05-25
Accepted:	2022-05-25

Scientific Editor: Novella Guidi

Transaction Report:

January 24, 2022

Re: Life Science Alliance manuscript #LSA-2021-01299-T

Professor Jui-Hung Hung
National Chiao Tung University
Department of Biological Science and Technology
75 Bo-Ai Street
Hsin-Chu 300
Taiwan

Dear Dr. Hung,

Thank you for submitting your manuscript entitled "Estimating Intraclonal Heterogeneity and Subpopulation Changes from Perturbational Bulk Gene Expression Profiles in LINCS L1000 CMap by Premnas" to Life Science Alliance. The manuscript was assessed by expert reviewers, whose comments are appended to this letter. We, thus, encourage you to submit a revised version of the manuscript back to LSA that responds to all of the reviewers' points.

Thank you for this interesting contribution to Life Science Alliance. We are looking forward to receiving your revised manuscript.

Sincerely,

B. MANUSCRIPT ORGANIZATION AND FORMATTING:

Reviewer #1 (Comments to the Authors (Required)):

Hsieh et al. describe a computational method, Premnas, which couples single cell RNAseq and cell type decomposition with perturbation gene expression profiles, to identify drug induced changes of subpopulations, and to propose drug combinations with toxic effect on all subpopulations. Both cell type decomposition and perturbation gene expression profiles are actively researched areas of current systems biology, so the topic of the manuscript can be interesting for the community. The authors used state-of-the-art methodologies for subtype identification (ACTIONet), decomposition (CIBERSORTx) and also the largest perturbation gene expression profile database (LINCS-L1000) for their analysis. However, I have several questions regarding the whole concept of the study, and also the validations.

Major:

1) The authors propose in Figure 1, that perturbation induced gene expression changes are not necessarily the consequence of perturbation induced signalling / pathway activity changes, but the compositional changes of subpopulations. Premnas predicts the subpopulation changes induced by perturbation. However, if a perturbation leads to a strong cytotoxic effect in one cell population, it is hard to imagine that it does not influence pathway activity, thus gene expression in the other subpopulations. Based on this, the two mechanisms (pathway activity and subpopulation changes) are observed parallel. Could the authors demonstrate that Premnas can identify subpopulation changes also on the background of "general" gene expression changes?

2) The authors suggest that if a perturbation leads to decreased proportion of a subpopulation, then the drug kills selectively this given subpopulation. However, it is possible, that drug treatment leads to transition of cells between different subpopulations (e.g.: changes in cell cycle). How can Premnas differentiate between these different mechanisms?

3) The authors use Premnas to identify drug "cocktails" (combinations), which have toxic effects on all subpopulations, thus these drug cocktails could be therapeutic effective by eliminating different cell populations together. This is an interesting hypothesis, but should be experimentally validated.

4) Alternatively, the already cited Ben-David et al. Nature article shows the different drug sensitivity of different (transcriptionally characterised) MCF7 clones. Could the authors validate their method using this dataset? I.e. using LINCS signatures, they can identify subpopulation selective drugs, and test whether the drug sensitivity in the Ben-David et al. dataset corresponds to this (by inferring the subpopulation component of different MCF7 clones)? Or could the authors use any other drug sensitivity dataset to validate their proposed method?

Reviewer #2 (Comments to the Authors (Required)):

In this manuscript, the authors present a novel approach, Premnas, to characterize and quantify cell subpopulations in large scale bulk transcriptomics datasets. Their framework makes use of different tools designed to deconvolute and map underlying subpopulations from RNAseq experiments. This is motivated to enhance the analysis of the Connectivity Map gene expression profiles, before and after drug perturbations, to characterize clonal heterogeneity and identify drug-resistant cellular populations.

I commend the authors for their work, which is well performed and structured, with clear aims and well thought validation examples. The main aim addresses important and current challenges in the field - impact of clonal heterogeneity in the emergence of drug resistant persister cells in cancer - although I have some significant concerns on the robustness of the results.

Major comments:

1. The main application of Premnas is focused on the Connectivity Map initiative which uses a reduced experimental representation of the transcriptome, i.e. L1000 assays measure only 1000 landmark representative transcripts, to trade for increased scalability in terms of samples screened. I am concerned that it is not possible to robustly infer cell subpopulations using bulk L1000 screens. The authors did not show satisfactory convincing evidence to address this nor commented on this limitation of the L1000 screens. Moreover, some of the methods the authors use in their framework are designed and benchmarked for RNAseq experiments. These methodological and technological aspects should be analysed, benchmarked and discussed more carefully in order to support the manuscript results and conclusions.

2. In the analysis of the 12 bulk GEPs of MCF-7 treated with FDI-6 across different time points, can the authors comment on the seemingly erratic behaviour of subpopulation 1? Subpopulation 1 is estimated to be completely inhibited after treatment in the first two time points (3h and 6h) but it is again detected at later time point (9h)? How does this reflect on the error margins of the subpopulation estimations of Premnas? Have the authors assessed the confidence interval of the predictions, e.g. bootstrapping?

3. Similarly to the previous point, on the LINCS L1000 bulk GEPs application Figure 5, how do the authors justify that higher drug doses, in contrast to lower doses, increases subpopulation representation (e.g. UNC-0638 and Gemcitabine)?

4. Cell heterogeneity and genetic drift are indeed important variables to be taken into account, particularly when considering cell lines. Although, I could not understand if Premnas subpopulation estimates indeed support this hypothesis across the L1000 datasets, i.e. if there is indeed large subpopulation diversity. I missed having some overall statistics of the subpopulations identified across the LINCS L1000 bulk GEPs and their frequency.

Minor comments:

1. "We colored the MCF-7 cells used for the previous subpopulation identification based on the expression of the two reported marker genes of PA cells (i.e., CD44 and CLDN1) and discovered that most of the cells expressing a higher degree of these marker genes tended to aggregate in subpopulation 2, 4, and 9 in the UMAP plot (Fig. S8)." It is difficult to visualize this from Fig S8. Could the authors improve contrasting between the colored cells from the background cells (in black) and provide the subpopulations cluster annotation? Also a quantification of the number of colored cells that fall in each cluster would be a better quantitative representation to support this claim.

2. Error in Figure 3e, cluster 8, percentage should be 11.1% (instead of 1.1%).

3. Typo in Figure 5b, "Gemcitabi" should read "Gemcitabine".

4. Supplementary figures are not referenced in the main text in ascending order, for example, from Fig S1 jumps to Fig S3-4, and from Fig S4 jumps to Fig S7.

5. Page 12: "that involves in" should read "involved in".

6. Text formatting problem in page 16

=====

Reviewer #1

Comment 1

The authors propose in Figure 1, that perturbation induced gene expression changes are not necessarily the consequence of perturbation induced signaling/pathway activity changes, but the compositional changes of subpopulations. Premnas predicts the subpopulation changes induced by perturbation. However, if a perturbation leads to a strong cytotoxic effect in one cell population, it is hard to imagine that it does not influence pathway activity, thus gene expression in the other subpopulations. Based on this, the two mechanisms (pathway activity and subpopulation changes) are observed parallel. Could the authors demonstrate that Premnas can identify subpopulation changes also on the background of "general" gene expression changes?

Reply:

We thank the reviewer for acknowledging the fact that there are both pathway and subpopulation changes involved in the response against perturbation in cell lines and for raising the relevant request. If we understand correctly, the reviewer was asking whether Premnas takes general gene expression changes into account while estimating subpopulation changes. In fact, Premnes uses CIBERSORTx to deduce the subpopulation composition based on the scRNA-seq derived

subpopulation characteristics, which are the specific signatures that can distinguish subpopulations, so the general gene expression changes of non-signature genes are not going to affect the estimation.

Just like the traditional CMap interpretation assumes no changes in subpopulation composition, we make the assumption in Premnes (i.e., Assumption 2 in the **Rationale**) that there exist invariant subpopulation characteristics to represent subpopulations before and after perturbation, so that the fluctuation of expression of these subpopulation characteristics can be solely explained by changes in subpopulation composition. Indeed, it is possible that the subpopulation characteristics can change upon perturbation and invalidate the role of signatures for indicating specific subpopulations.

That is to say, these two approaches (the conventional approach and ours) make their own assumptions and therefore have flaws to describe how cells respond to perturbation. Such compromise has to be made, since it is clearly much harder, if not impossible, to devise a model without needing such an assumption. Researchers should understand the limitation of the approaches and explain the results of either model with caution and thereby benefit from interpreting things from two parallel perspectives.

In order to avoid misleading the readers, we made the following change in corresponding paragraphs as follows:

In Results/Rationale:

“Assumption 2. Cells of the same subpopulation should collectively share invariant subpopulation characteristics and each subpopulation can be distinguished by its unique subpopulation characteristics despite perturbations.”

“Performing digital cytometry. Once the underlying subpopulations were identified, the most straightforward way to estimate their abundance in bulk samples is by conducting a simple linear regression modeling the relationship between the bulk GEP and subpopulation characteristics. ... CIBERSORTx is capable of adjusting the matrix of subpopulation characteristics derived from the scRNA-seq GEPs while decomposing the query bulk GEPs into the distribution of cell subpopulations with support vector regression.”

In Discussion.

“The logical basis of Premnas relies on the assumption that there are invariant subpopulation characteristics to represent each subpopulation so that the fluctuation of expression of these subpopulation characteristics can be solely explained by changes in subpopulation composition. However, in practice, the inferred gene signatures can be the mixed consequences of the subpopulation and function changes, therefore violating the assumption. As a result, it is possible that the subpopulation changes reported by Premnas can be due to cells changing their behaviors and acting like some other subpopulations upon a treatment. Unfortunately, it is pretty unlikely such a difference

can be distinguished from the information given in the bulk GEPs in the current setting. It is strongly recommended to always refer to the DE genes or enriched functions associated with the major archetypes of the affected subpopulations and thereby interpret the results also from the function perspective. It is important to keep open to alternative explanations of the results."

Comment 2

The authors suggest that if a perturbation leads to decreased proportion of a subpopulation, then the drug kills selectively this given subpopulation. However, it is possible, that drug treatment leads to transition of cells between different subpopulations (e.g.: changes in cell cycle). How can Premnas differentiate between these different mechanisms?

Reply:

Thanks for the question, it is a valid concern. This is also the main reason we deliberately include some normalization and preprocessing steps in Premnas to avoid learning the subpopulation characteristics that indicate irrelevant states due to technical and biological noise. After the procedure (see *Removal of biological or technical noise* in **Methods**), as shown in **Fig. S10b**, cells in different cell cycle phases distributed evenly in the embedding space, therefore the clustering would not identify subpopulations linking to cell cycle phases. As the result, general gene changes due to the cell cycle would not affect the identification of subpopulations.

However, indeed, we cannot rule out the possibility that a type of cells can present characteristics of other types of cells after drug treatment, but it is beyond the capacity of Premnas, please refer to our reply to Comment 1 above for an explanation.

Comment 3

The authors use Premnas to identify drug "cocktails" (combinations), which have toxic effects on all subpopulations, thus these drug cocktails could be therapeutic effective by eliminating different cell populations together. This is an interesting hypothesis, but should be experimentally validated.

Reply:

Thanks for the comment, we are glad to know that the reviewer liked the idea. We agree that more experiments are needed to further validate the effects of drug cocktails found by Premnas. However, as a dry lab, performing wet-lab experiments is beyond our capacity, and our intention of proposing the drug cocktail idea was only to provide a possible application of Premnas and we had tried our best to

collect evidence to support the findings. After discussing with the editor, the editor suggested us tuning down the statement about the cocktails and moving it to **Discussion** in the manuscript. We, therefore, removed the statements about the drug cocktail from **Abstract** and **Conclusions**, and move the entire drug cocktail section to **Discussion**:

In Abstract:

“By recovering the information of subpopulation changes upon perturbation, the potentials of drug-resistant/susceptible subpopulations with CMap L1000 were further explored and examined”

In Conclusions:

“We applied Premnas to MCF-7 cell line data and identified ten cell subpopulations. We found consistent experimental evidence to support the classification. After dissecting the effects of thousands of perturbations on MCF-7 cells from the bulk profiling assays curated in the LINCS CMap, we further discovered the most resistant subpopulation among MCF-7 cells and associated its characteristics to the known PA cells. The result suggested that Premnas can be applied to perturbation datasets to reveal intraclonal/intratatumoral heterogeneity and provides a new dimension of interpreting signatures and connectivity.”

In Discussion:

“After getting the drug susceptibility and treatment consistency of all PCT pairs of LINCS L1000 MCF-7 we came up with a greedy search strategy for suggesting a minimal therapeutics combination (i.e., a cocktail therapy) by aggregating perturbagens that kill specific subpopulations, thereby no subpopulation could survive after the treatment.”...followed by the original section A greedy search strategy of suggesting cocktails for suppressing breast tumor growth using LINCS L1000 CMap)”

And

“We did not carry out further experiments to verify the effectiveness of the drug cocktail but there are many studies that have already proved the anti-tumor activities of each selected compound, supporting the feasibility of this treatment combination.”

Comment 4.

Alternatively, the already cited Ben-David et al. Nature article shows the different drug sensitivity of different (transcriptionally characterised) MCF7 clones. Could the authors validate their method using this dataset? I.e., using LINCS signatures, they can identify subpopulation selective drugs, and test whether the drug sensitivity in the Ben-David et al. dataset corresponds to this (by inferring the subpopulation component of different MCF7 clones)? Or could the authors use any other drug sensitivity dataset to validate their proposed method?

Reply:

We thank the reviewer for the suggestion. In fact, we were thinking the same for using perturbational datasets (before and after a treatment) to support our method. However, in the Nature article by Ben-David et al., the authors showed the different drug sensitivity of 27 different MCF7 strains by cell viability, and did not provide suitable data for estimating the subpopulation.

Actually, in the manuscript, we did take other independent datasets to validate our proposed method. We used FDI-6 data (GSE58626) to confirm that Premnas could successfully detect the drug sensitivity between cells/subpopulations (the details are mentioned in **Drug-susceptible subpopulation inferred from bulk GEPs reflects drug-induced pathway**). Moreover, we also showed that the characteristics of the drug-resistant cell subpopulation (subpopulation 2) recognized in MCF-7 by Premnas are highly similar to that of the known “pre-adapted cells” with resistance against the endocrine therapy (Hong et al. 2019) (see **Fig. 3e**).

Once again, we thank Reviewer 1 for all the kind comments and suggestions.

=====

Reviewer #2

Comment 1.

The main application of Premnas is focused on the Connectivity Map initiative which uses a reduced experimental representation of the transcriptome, i.e. L1000 assays measure only 1000 landmark representative transcripts, to trade for increased scalability in terms of samples screened. I am concerned that it is not possible to robustly infer cell subpopulations using bulk L1000 screens. The authors did not show satisfactory convincing evidence to address this nor commented on this limitation of the L1000 screens.

Reply:

We thank the reviewer for bringing up the concern toward whether 1000 landmark genes are sufficient to infer subpopulation. In fact, in a typical scRNA-seq analysis pipeline, it is common to perform gene filtering that keeps only about 1 or 2 thousand most variable genes in the GEP. The GEP then undergoes the PCA transformation and only maybe 10 or 20 top principal components are used for

subpopulation clustering. That is, when identifying subpopulation, the representativeness of genes is more important than the number of genes, it is common to use only representative transcripts to characterize a subpopulation. Moreover, according to the Cell article by Subramanian et al., the L1000 genes were able to recover 82% of the information of the transcriptome of 12,031 Affymetrix HGU133A expression profiles. A strong degree of similarity of profiles of L1000 and RNA-seq was also shown in the comparison of 3,176 samples. In addition, we have used many independent datasets to support the correctness of our decomposition. We think it is reasonable to assume that 1000 landmark genes are sufficient for the task. We hope the reviewer can agree with us after the explanation.

Comment 2.

Moreover, some of the methods the authors use in their framework are designed and benchmarked for RNAseq experiments. These methodological and technological aspects should be analysed, benchmarked and discussed more carefully in order to support the manuscript results and conclusions.

Reply:

We thank the reviewer for pointing out the technical biases introduced by different platforms are crucial. In fact, this issue has been addressed by the digital cytometry method, CIBERSORTx, in which a sophisticated normalization process has been included to deal with bulk profiles from microarray and RNA-seq. Premnas relies on CIBERSORTx to tackle platform biases. We now emphasize the importance of data normalization and the possibility that the learnt subpopulation characteristics are not representative for bulk GEPs in **Discussion** to clarify this:

In Discussion:

"The differences between profiling technologies place a difficulty in estimating subpopulation distribution in bulk samples. CIBERSORTx (S-mode) reduced the technical variation in gene expression by using an artificial mixture to help tune the signature matrix (see Materials and Methods). Furthermore, the bulk GEPs we encountered were largely conducted by the L1000 and RNA-seq, and they were designed to quantify different gene sets. That is, it is possible that some genes involved in the learning of subpopulation characteristics do not present in bulk GEPs. Since CIBERSORTx is a marker gene-based decomposition approach, the calculation could depend on some of those missing genes, thereby compromising accuracy."

To further address the reviewer's concern that our methods were only benchmarked for RNA-seq experiments, we then add more experiments to validate the feasibility of using Premnas for deconvoluting microarray data. The microarray data of PBMCs from 10 humans were downloaded from the GEO web site with the

accession number GSE106898, and the deconvolution outputs were examined as we had done for the PBMC bulk RNA-seq dataset (see PBMC verification in Supplementary Materials). The experiment results were shown in the new **Fig. S2**, in which the correlation coefficient between the composition estimations via the digital cytometry and the ground truth was also high ($r=0.8$ by Pearson correlation coefficient), suggesting that Premnas can estimate the distribution of cell subpopulations in microarray correctly. The new figure and corresponding texts are now read as follows:

In **Supplementary Materials** (new figure added):

Figure S2 - PBMC microarray samples decomposition. Using the cell subpopulations signature produced by Premnas to deconvolute PBMC microarray datasets. The microarray data of PBMCs from 10 humans were downloaded from the GEO website with the accession number GSE106898. (a) Scatter plot of subpopulation composition with x-axis as the ground truth and y-axis as the estimation. Each point represents a specific cell type of one PBMC microarray sample. (b) Bar plot shows the decomposition performance on cell-type level with Pearson correlation coefficient.

In Materials and Methods:

“Microarray data: The microarray data of PBMCs from 10 humans was downloaded from the GEO website with the accession number GSE106898. The expression data was quantile normalized and the probe IDs were transformed into gene names accordingly.”

In Results/Validation:

“The Pearson correlation coefficient between the composition estimations via the digital cytometry based on the ad hoc subpopulation characteristics and the ground truth composition directly assessed by flow cytometry was high ($r=0.835$) (see Fig. S1c and d). Moreover, in addition to the bulk RNA-seq, we also performed the deconvolution validation on the microarray (see Fig. S2) platform. The estimation based on microarray also showed a high correlation with the ground truth ($r=0.80$ by Pearson correlation coefficient). These results suggesting that Premnas can discover the unspecified subpopulation from scRNA-seq data and estimate the distribution of cell subpopulations in bulk samples correctly.”

Comment 3.

In the analysis of the 12 bulk GEPs of MCF-7 treated with FDI-6 across different time points, can the authors comment on the seemingly erratic behaviors of subpopulation 1? Subpopulation 1 is estimated to be completely inhibited after treatment in the first two time points (3h and 6h) but it is again detected at later time point (9h)? How does this reflect on the error margins of the subpopulation estimations of Premnas? Have the authors assessed the confidence interval of the predictions, e.g. bootstrapping?

Reply:

We thank the reviewer for pointing out the seemingly erratic behaviors of subpopulation 1 in the FDI-6 treated samples and suggesting an assessment of the confidence interval by bootstrapping. Before we can attribute the seemingly erratic behavior to error in prediction, we first examined the error margins of subpopulation estimations of Premnas according to the suggestion. In fact, there is a sampling parameter in CIBERSORTx, which could help us get an idea of the robustness of the prediction. We first tested five sampling fractions (0.2, 0.3, 0.4, 0.5 and 0.6, which represent the sampling proportion from all single-cell GEPs that CIBERSORTx used to build the single-cell signature matrix) and examined the deconvolution results of 4 PBMC bulk samples between these five fractions. The results shown in the new **Fig. S12** indicated that the five subpopulation estimations followed a similar trend with some fluctuations. Then we did bootstrapping to see the range of confidence interval, and we saw the similar convergent trend among all estimations (new **Fig. S13**). However, we did observe some subpopulations were more prone to have outliers leading to large confidence intervals.

We think the large confidence interval can be attributed to the nature of the high variance in the scRNA-seq GEPs, which can also be manifested in how each cell is projected in embedding space. For example, as showed in Fig. 3a, for the MCF-7 data, we can always see cells of one subpopulation blend into other subpopulations. In other words, there are many cells show characteristics of more than one subpopulation. Take subpopulation 1 as an example, its characteristics are somehow similar to subpopulation 3, 6, and 7 (see new **Fig. S5**).

Another cause of large confidence intervals can be stemmed from the assumptions taken in Premnas. Just like the traditional CMap interpretation assumes no changes in subpopulation composition, we make the assumption in Premnes (i.e., Assumption 2 in the **Rationale**) that there exist invariant subpopulation characteristics to represent subpopulations before and after perturbation, so that the fluctuation of expression of these subpopulation characteristics can be solely

explained by changes in subpopulation composition. Indeed, it is possible that the subpopulation characteristics can change upon perturbation and invalidate the role of signatures for indicating specific subpopulations.

That is to say, these two approaches (the conventional approach and ours) make their own assumptions and therefore have flaws to describe how cells respond to perturbation. Such compromise has to be made, since it is clearly much harder, if not impossible, to devise a model without needing such an assumption. Researchers should understand the limitation of the approaches and explain the results of either model with caution and thereby benefit from interpreting things from two parallel perspectives.

Therefore, we think it is possible that changes of pathways upon FDI-6 treatment could lead to cells of subpopulation 3, 6, and 7 to behave more like subpopulation 1, thereby violating the Assumption 2 and causing aberrant prediction. The seemingly erratic behaviors of subpopulation 1 in the FDI-6 treated samples, were caused by misinterpreting cells from subpopulation 3 as subpopulation 1 (since subpopulation 6 and 7 were likely completely inhibited as well).

In order to make our point clearer, we made the following change in corresponding paragraphs as follows:

In Results/Rationale:

“Assumption 2. Cells of the same subpopulation should collectively share invariant subpopulation characteristics and each subpopulation can be distinguished by its unique subpopulation characteristics despite perturbations.”

“Performing digital cytometry. Once the underlying subpopulations were identified, the most straightforward way to estimate their abundance in bulk samples is by conducting a simple linear regression modeling the relationship between the bulk GEP and subpopulation characteristics. ... CIBERSORTx is capable of adjusting the matrix of subpopulation characteristics derived from the scRNA-seq GEPs while decomposing the query bulk GEPs into the distribution of cell subpopulations with support vector regression.”

In Discussion:

“The logical basis of Premnas relies on the assumption that there are invariant subpopulation characteristics to represent each subpopulation so that the fluctuation of expression of these subpopulation characteristics can be solely explained by changes in subpopulation composition. However, in practice, the inferred gene signatures can be the mixed consequences of the subpopulation and function changes, therefore violating the assumption. As a result, it is possible that the subpopulation changes reported by Premnas can be due to cells changing their behaviors and acting like some other subpopulations upon a treatment. Unfortunately, it is pretty unlikely such a difference can be distinguished from the information given in the bulk GEPs in the current setting. It is strongly recommended to always refer to the DE genes or enriched functions associated with the major archetypes of the affected subpopulations and thereby interpret

the results also from the function perspective. It is important to keep open to alternative explanations of the results.”

In Materials and Methods:

“We also performed some sampling experiments from the PBMC datasets to examine the robustness of decomposition by CIBERSORTx (see Fig.S12-13).”

In Supplementary Materials (new Figures added):

Figure S5 - Overlapping proportions of highly expressed genes between subpopulations. The heatmap shows the overlapping proportion of the top 50 highly expressed genes between two subpopulations. The proportion was count by: $\frac{\text{Union} (\text{top 50 genes of A subpopulation} , \text{top 50 genes of B subpopulation})}{\text{Intersection} (\text{top 50 genes of A subpopulation} , \text{top 50 genes of B subpopulation})}$.

Figure S12 - Robustness of different single-cell signature matrix construction parameters. CIBERSORTx constructs the single-cell signature matrix by sampling a proportion of all single-cell GEPs using random sampling without replacement (default sampling fraction = 0.5). To examine the robustness of the digital cytometry approach and the stability of the single-cell signature, we tested five sampling fractions (0.2, 0.3, 0.4, 0.5 and 0.6) and visualized the deconvolution results of 4 PBMC bulk samples between these fractions.

Figure S13 - Robustness of subpopulation estimations by CIBERSORTx. We reran the CIBERSORTx ten times with fixed parameters and evaluated the consistency between these subpopulation estimation results in four PBMC data, in which bulk data was the same as Fig. S12. The parameters were set with default values and the S-mode correction was used in CIBERSORTx. The figure shows that the proportion of eight PBMC subpopulations wasn't changed substantially, implying the robustness of subpopulation estimations by CIBERSORTx.

Comment 4.

Similarly to the previous point, on the LINCS L1000 bulk GEPs application Figure 5, how do the authors justify that higher drug doses, in contrast to lower doses, increases subpopulation representation (e.g. UNC-0638 and Gemcitabine)?

Reply:

We appreciate the carefulness of the reviewer to raise more seemingly erratic cases. We think these unexpected behaviors could all be linked to the limitation of the assumption taken by Premnas. Please refer to our reply to Comment 3 above for an explanation.

Comment 5.

Cell heterogeneity and genetic drift are indeed important variables to be taken into account, particularly when considering cell lines. Although, I could not understand if Premnas subpopulation estimates indeed support this hypothesis across the L1000 datasets, i.e. if there is indeed large subpopulation diversity. I missed having some overall statistics of the subpopulations identified across the LINCS L1000 bulk GEPs and their frequency.

Reply:

We are glad that the reviewer agrees with us that cell heterogeneity and genetic drift are both important factors to consider. To confirm that there is indeed large subpopulation diversity across the LINCS L1000 bulk GEPs, we summarized the subpopulation proportion distributions of 39,710 LINCS L1000 bulk data for MCF7 as shown in the figure below.

As we can see from the figure, the proportions of some subpopulations (especially subpopulations 2 and 5, i.e., S2 and S5) after treatment are pretty varied. And interestingly, subpopulation 2 tends to take up the largest proportion after treatment on average, supporting our finding that subpopulation 2 is drug-resistant.

Once again, we thank Reviewer 2 for all the kind comments and suggestions.

May 18, 2022

RE: Life Science Alliance Manuscript #LSA-2021-01299-TR

Prof. Jui-Hung Hung
National Chiao Tung University
Department of Biological Science and Technology
75 Bo-Ai Street
Hsin-Chu 300
Taiwan

Dear Dr. Hung,

Thank you for submitting your revised manuscript entitled "Estimating Intraclonal Heterogeneity and Subpopulation Changes from Bulk Gene Expression Profiles in LINCS L1000 CMap". We would be happy to publish your paper in Life Science Alliance pending final revisions necessary to meet our formatting guidelines.

- please add the Twitter handle of your host institute/organization as well as your own or/and one of the authors in our system
- please add the legends for the Tables to the main manuscript text
- please add callouts for Figure 3C & 3E and Figure S1B, Figure S4C, to your main manuscript text

A. FINAL FILES:

B. MANUSCRIPT ORGANIZATION AND FORMATTING:

**Submission of a paper that does not conform to Life Science Alliance guidelines will delay the acceptance of your

manuscript.**

The license to publish form must be signed before your manuscript can be sent to production. A link to the electronic license to publish form will be sent to the corresponding author only. Please take a moment to check your funder requirements.

Sincerely,

Reviewer #1 (Comments to the Authors (Required)):

I think the authors answered most of my questions, and moved some experimental not verified parts from results to discussion.

I have no remaining questions.

Reviewer #2 (Comments to the Authors (Required)):

I thank the authors for their thorough responses and changes made to manuscript. While I appreciate their arguments in support of the L1000 platform, some practical benchmarks comparing with RNAseq would have been preferable. Nonetheless, my concerns were satisfactorily addressed, and I believe this manuscript is suitable for publication.

May 25, 2022

RE: Life Science Alliance Manuscript #LSA-2021-01299-TRR

Prof. Jui-Hung Hung
National Chiao Tung University
Department of Biological Science and Technology
75 Bo-Ai Street
Hsin-Chu 300
Taiwan

Dear Dr. Hung,

Thank you for submitting your Research Article entitled "Estimating Intraclonal Heterogeneity and Subpopulation Changes from Bulk Expression Profiles in CMap". It is a pleasure to let you know that your manuscript is now accepted for publication in Life Science Alliance. Congratulations on this interesting work.

DISTRIBUTION OF MATERIALS:

Again, congratulations on a very nice paper. I hope you found the review process to be constructive and are pleased with how the manuscript was handled editorially. We look forward to future exciting submissions from your lab.

Sincerely,
